# The coupling coordination characteristics of China's health production efficiency and new urbanization and its influencing factors

**Haili Zhao, Fang Zhang** *, **Yuhan Du, Jialiang Li, Minghui Wu**

College of Geography and Environment Science, Northwest Normal University, Lanzhou, 730070, China

* 2021212840@nwnu.edu.cn

## Abstract

Urbanization leads to dramatic changes in habitat quality, which significantly affects population health. Research on the coupling coordination relationship between new urbanization and health production efficiency is conducive to improving residents' well-being and urban sustainable development. In this article, we adopted the super-efficient SBM model and entropy value method separately to evaluate the spatiotemporal variation characteristics of health production efficiency and new urbanization in China. Then, we used the coupling coordination degree model to investigate the interactive coercing relationship between new urbanization and health production efficiency. Finally, the panel Tobit model is used to analyze the factors influencing the coupled coordination of the two systems. The results showed that the new urbanization levels of 31 provinces in China have all steadily increased from 2003 to 2018. Health production efficiency exhibited a fluctuating but increasing trend, and its regional differences are gradually narrowing. Health production efficiency and new urbanization have developed in a more coordinated direction, with a spatial pattern of "high in the southeast and low in the northwest." Meanwhile, the relative development characteristics between the two systems have constantly changed, from the new urbanization lagged type to the two systems synchronized type and the health production efficiency lagged type. Population density, economic development level, government financial investment, and government health investment positively impact the coupling coordination degree of the two systems. In comparison, individual health investment harms the harmonization of the two systems.

## 1 Introduction

New urbanization and the Healthy China strategy are essential national development strategies in China. The proposal for new urbanization reflects, adjusts, and explores China's urbanization road. It is an urbanization with more Chinese characteristics and emphasizes human-land coordination and sustainability [1]. Health is the eternal pursuit of human survival and development. The idea of putting the protection of people's health in the priority development position was put forward at the 20th National Congress of the Communist Party of China. The process of new urbanization is a process of interaction and coordination between people and

**Funding:** This work was supported by the National Natural Science Foundation of China (grant number

42361036). The funders had no role in study design, data collection and analysis, decision to publish, or preparation of the manuscript.

**Competing interests:** The authors have declared that no competing interests exist.

land. As one of the results of the interaction between people and land, population health has an obvious interaction between the two. On the one hand, new urbanization has changed the built environment. which has positive and negative impacts on population health. The positive impact is that it has brought about more convenient living conditions and service facilities, shaping the development of human health in a sustainable direction [2]. The negative impact is that air pollution and climate change resulting from urbanization increase the risk of humans contracting infectious diseases [3]. Studies have shown that the incidence of diseases such as obesity, respiratory diseases, cardiovascular diseases, and mental health are closely related to the negative impacts of urbanization on the environment [4–7]. On the other hand, the new urbanization of population health also has both positive and negative impacts, the positive impact being that the continuous improvement of public health not only increases labor productivity and improves the accumulation of human capital but also helps to promote high-quality economic growth through the promotion of health consumption and investment [8]. The negative impact is that unhealthy and unhealthy will increase government and individual expenditures on health, thereby increasing socioeconomic pressures. In general, new urbanization and health production efficiency have a conflicting relationship. It is essential to explore the interaction between the two to enhance the quality of urbanization, increase the well-being of urban residents, and further achieve the sustainable development of cities.

Health production efficiency is often used to measure the health production capacity of each region's medical and health system, which is a crucial factor restricting health output. In recent years, with the promotion of the Healthy China strategy and the development of the new urbanization of "people-oriented," investment in the health field has been increasing. However, the level of health output is restricted by both health input and health production efficiency. Blindly increasing health input while ignoring health production efficiency will only cause a waste of resources [9]. Therefore, only the synchronous growth of health production efficiency and new urbanization can better promote healthy and coordinated economic and social development. Domestic scholars' research on health production efficiency mainly focuses on three aspects: efficiency measurement, influencing factors, and regional differences analysis. Among them, Zhang et al. considered the regional health system as a health production decision-making unit and evaluated the efficiency of health production in each region of China through the data envelopment analysis method (DEA) [10]. Yu et al. conducted a full-factor analysis of health production efficiency from five aspects: economy, population, education, health, and government attention [11]. The study by Han et al. showed that population density, education level, per capita GDP, fiscal decentralization, and medical system reform are important reasons for regional differences in health production efficiency [12]. In addition to the traditional "efficiency-influencing factors" two-step research paradigm, foreign scholars have also considered the correlation between efficiency and equity [13], efficiency and medical service quality [14], and efficiency and public health policy [15].

At present, the research on new urbanization in academia covers the concept of new urbanization [1], evaluation system research [16], coupling coordination research [17], regional development drive [18], basic public services [19], and development strategies. Among them, Li et al. proposed that the new urbanization should essentially emphasize the coordination of urban and rural areas, coordination between landscape urbanization and population urbanization, and coordination between residential urbanization and public service urbanization [20]. Based on the concept of humanistic development, Chen et al. emphasized that new urbanization should gradually realize the transformation from "population urbanization" to "human urbanization" and strive to improve the accessibility of medical, educational, cultural, and sports services for urban disadvantaged groups and migrant workers, and to promote the equalization of basic public services in urban and rural areas [1].

Given the above considerations, the existing literature has comprehensively studied new urbanization and health production efficiency. However, new urbanization and health production efficiency, as two systems, have mutual influences and constraints. The current studies mainly focus on the one-way influence mechanism of the former on the latter [9–12]. Few studies have been conducted from the perspective of health production efficiency, researching the coordination between new urbanization and health production efficiency. This study mainly revolves around two scientific problems: (1) how does the relationship between new urbanization and health production efficiency evolve in China? (2) what factors affect the coordinated development of the two systems? To solve these scientific problems, we based on health production theory and used the coupled coordination degree model to analyze the interrelationship between new urbanization and health production efficiency from 2003–2018. Specifically, our research objectives were to (1) explore the spatiotemporal characteristics of health production efficiency in Chinese regions from 2003–2018, (2) evaluate the new urbanization levels in Chinese provinces during this period from dimensions of demography, space, economy, and society, (3) reveal the coupling coordination relationship between new urbanization and health production efficiency, and (4) explore the factors affecting the coordinated development of the two system.

## 2 Materials and methods

### 2.1 Index system and research methods

**2.1.1 Construction of evaluation model for new urbanization.** The measurement of urbanization level mainly has a single and comprehensive index method. Since a single index of population urbanization does not reflect the comprehensive development level of urbanization, scholars mainly adopt the comprehensive indicator method to construct it from the connotation of urbanization, including at least population urbanization, spatial urbanization, economic urbanization, and social urbanization. At present, with the introduction of various policies on new urbanization, indicators such as industry, green, and urban-rural integration have also been covered. Referring to the existing research results on new urbanization [21–25], this paper constructs a new urbanization evaluation index system from six dimensions: population urbanization, economic urbanization, spatial urbanization, social urbanization, green urbanization, and urban-rural integration (Table 1).

In this paper, we use the panel entropy method to calculate the new urbanization level of each region from 2003 to 2018, which is performed in the software Stata17.0. The entropy method primarily utilizes calculations of the original data of indicators to assess the randomness and degree of disorder of a particular phenomenon or indicator information. Based on this, it determines the impact of the indicator on the overall comprehensive evaluation, thereby avoiding the bias introduced by subjective factors to some extent. In order to overcome the limitations of traditional entropy methods that could only handle cross-sectional data and could not compare different years, this paper adopts the panel entropy method improved by Yang L and Sun Z [26]. The specific steps are as follows:

1. Indicator selection. Let there be $r$ years, $n$ provinces, and $m$ indicators, then $x_{ijk}$ denotes the value of the $k$-th indicator in the $j$-th province in the $i$-th year.

2. Indicator normalization.

   Normalization of positive indicators: $x'_{ijk} = \frac{x_{ijk} - min\{x_{ijk}\}}{max\{x_{ijk}\} - min\{x_{ijk}\}}$

   Normalization of negative indicators: $x'_{ijk} = \frac{max\{x_{ijk}\} - x_{ijk}}{max\{x_{ijk}\} - min\{x_{ijk}\}}$

**Table 1. Evaluation index system of new urbanization level.**

| System layer | Grade I index | Grade II index | Description of index | index weight |
|---|---|---|---|---|
| New urbanization | Population | Population urbanization rate | The proportion of the urban population | 0.0612 |
| | | The proportion of urban employment | The proportion of employees in secondary and tertiary industries | 0.0446 |
| | Economic | Economic development level | Per capita GDP | 0.1507 |
| | | Urban industrial structure | The proportion of output value of secondary and tertiary industries | 0.0171 |
| | Space | Urban road carrying capacity | Per capita urban road area | 0.0691 |
| | | Land carrying capacity | Per capita built-up area | 0.0871 |
| | Social | Public service level | Public transport vehicles per 10,000 people | 0.0731 |
| | | Level of urban civilization | Population with a college degree or above | 0.1116 |
| | | Living standard | Private car ownership | 0.3106 |
| | Green | City livable level | The green coverage rate of built-up area | 0.0100 |
| | | Domestic pollution treatment | Treatment rate of domestic sewage | 0.0397 |
| | Urban-rural integration | Urban-rural consumption gap | The ratio of urban-rural per capita consumption expenditure | 0.0049 |
| | | Urban-rural income gap | The ratio of urban-rural per capita disposable income | 0.0202 |

3. Calculation of indicator weights: $y_{ijk} = x'_{ijk} / \sum_{i=1}^{r} \sum_{j=1}^{n} x'_{ijk}$

4. Calculation of entropy value for the $k$-th indicator:

$$e_k = -\frac{1}{\theta} \sum_{i=1}^{r} \sum_{j=1}^{n} y_{ijk} \ln\left(y_{ijk}\right)$$

where: $\theta = \ln(rn), e_k \in [0,1]$

5. Calculation of information utility value for the $k$-th indicator: $g_k = 1 - e_k$

6. Calculation of weights for the $k$-th indicator: $w_k = g_k / \sum_{k=1}^{m} g_k$

7. Calculation of the comprehensive score for new urbanization in each province annually: $h_{ij} = \sum_{k=1}^{m} w_k x'_{ijk}$

**2.1.2 Health production efficiency evaluation model.** The health production efficiency evaluation model is based on the health production theory proposed by Grossman in 1972 [27]. According to this theory, health is a "commodity" with dual attributes of consumption and investment. The health status of an individual at birth is the initial stock of health owned by the individual, and this stock depreciates at a certain depreciation rate in a certain period. The increase of health input will increase the stock of health, but when the stock of health is low to a certain level, death will come. This theory describes the healthy production process from an economic perspective for the first time and provides a theoretical basis for the study of health production efficiency. In this theory, health inputs are divided into two categories: healthcare inputs and other socioeconomic inputs. Since sufficient data on other social environment variables are not available, the treatment of social environment variables is not consistent, and some scholars believe that those variables affect health output indirectly, mainly through the ability and awareness of health inputs. Therefore, the health production efficiency model is simplified, and the regional medical and health system is considered a health

production decision unit. Hence, health production efficiency is the relative efficiency of transforming medical and health input into health output.

Medical and health input generally involves human, material, and financial resources. Among them, human input mainly refers to health technicians, and this paper uses the number of health technicians per 1000 population to characterize. Material input mainly refers to the number of beds and medical equipment in medical and health institutions, but because it is difficult to obtain complete and accurate medical equipment statistics, this paper uses the number of beds per 1,000 population to characterize material input; Financial inputs represent the investment of healthcare funds. In this study, three indicators are selected to measure the financial inputs of health production: per capita government health expenditure, per capita healthcare expenditure of urban residents, and per capita healthcare expenditure of rural residents. These indicators can not only differentiate between government and individual health investments but also reflect the disparities in health investments between urban and rural residents. In terms of health output, the existing literature mainly focuses on indicators such as per capita life expectancy, mortality, and disease incidence. Perinatal mortality, maternal mortality, and infectious disease incidence can describe health output from different perspectives, so they are widely used as health output variables. In this paper, the above three indicators are also selected as output variables and are positively transformed as the expected output.

Data envelopment analysis (DEA) is a commonly used non-parametric evaluation method for assessing the relative efficiency of decision-making units (DMUs) with multiple inputs and outputs. The principle of DEA is based on the formation of a production frontier by some DMUs' production behavior within a comparable set of DMUs. DMUs on the production frontier are considered technically efficient, while the technical efficiency of other DMUs is constructed based on distance functions relative to the production frontier. DEA can be divided into two categories: radial models represented by the CCR and BCC models and non-radial models represented by the SBM model. The super-efficiency SBM model proposed by Tone in 2002 can effectively handle unexpected outputs [28], address the "crowding" or "slack" phenomena of input and output variables, and further evaluate the efficiency differences among DMUs on the frontier, resolving the comparative evaluation issue when multiple DMUs are simultaneously efficient. Therefore, this study adopts the super-efficiency SBM model to measure the efficiency of health production. Since health production systems generally aim to maximize health output levels under given budget constraints, an output-oriented form is chosen to construct a current efficiency model. The specific calculation formula is as follows [29,30].

$$
\begin{cases}
\min_{\rho} = \dfrac{1 + \dfrac{1}{m}\sum_{i=1}^{m}\dfrac{s_i^-}{x_{ij}}}{1 - \dfrac{1}{q}\sum_{r=1}^{q}\dfrac{s_r^+}{y_{rj}}} \\[4ex]
\text{s.t.} \displaystyle\sum_{j=1,j\neq k}^{n} x_{ij}\gamma_i + s_i^- \leq x_{ij} \\[3ex]
\displaystyle\sum_{j=1,j\neq k}^{n} y_{rj}\gamma_j - s_r^+ \geq y_{rj};\ \gamma_j, s^+, s^- \geq 0
\end{cases}
\tag{1}
$$

Where i is the input indicators with a range of [1-m], r is the output indicators with a range of [1-q], ρ represents the health production efficiency of the evaluated DMU. $x_{ij}$ and $y_{rj}$ are the input and output values of j DMU, respectively. $s^+$ and $s^-$ represent the slack variables for input

and output, respectively. γ is the weight vector. The above algorithm is performed in the software MaxDEA.

**2.1.3 Coupling coordination model.** The coupling coordination model can explore the coupling and coordination relationships between two or more systems. Among them, the coupling degree model characterizes the degree of mutual influence between multiple systems. The larger the coupling degree, the stronger the relationship between the systems and the more orderly the development direction. This paper introduces this model to measure the interrelationship between health production efficiency and new urbanization. The specific formulas are as follows:

$$C = 2\left[\frac{U_1 \cdot U_2}{(U_1 + U_2)^2}\right]^{\frac{1}{2}} \tag{2}$$

Where $U_1$ and $U_2$ are the values of health production efficiency and new urbanization, respectively, and C is the coupling degree between the two, with a range of [0.1].

The calculation of the coupling coordination degree model is established based on the coupling degree model, which can truly reflect the synergistic effect of the two systems. So, this paper introduces the model to quantitatively analyze the temporal and spatial evolution characteristics of the coupled and coordinated development of health production efficiency and new urbanization. The specific model is as follows:

$$D = \sqrt{CT}; T = \alpha U_1 + \beta U_2 \tag{3}$$

Where α and β, respectively, are the undetermined coefficients of the two systems used to measure the share of each system. The author believes that the development of the two systems is equally important, so set α = β = 0.5. T is the comprehensive development index of the two systems, which reflects the overall coordination level of the two systems. D is the coupling coordination degree of the two systems with a range of [0.1]. According to the existing literature, this paper classifies the coupling coordination degree into ten levels: $0 < D \leq 0.1$ indicates extreme incoordination; $0.1 < D \leq 0.2$ indicates severe incoordination; $0.2 < D \leq 0.3$ indicates moderate incoordination; $0.3 < D \leq 0.4$ indicates mild incoordination; $0.4 < D \leq 0.5$ indicates on the verge of incoordination; $0.5 < D \leq 0.6$ indicates barely coordinated; $0.6 < D \leq 0.7$ indicates primary coordination; $0.7 < D \leq 0.8$ indicates moderate coordination; $0.8 < D \leq 0.9$ indicates good coordination; $0.9 < D \leq 1.0$ indicates high-quality coordination. The calculation of the coupling coordination degree is implemented in Excel and its spatial visualization was performed in ArcMap 10.2.

The coupling coordination model reflects the degree of coordinated development between health production efficiency and new urbanization. However, it does not reveal the relative development relationship between the two. Therefore, this paper introduces the relative development model to determine their relative development levels, with the formula as follows:

$$E = U_2/U_1 \tag{4}$$

In the formula, E is the relative development degree, $U_1$ is the standardized health production efficiency, and $U_2$ is the standardized level of new urbanization. When $0 < E \leq 0.8$, the level of urbanization lags behind health production efficiency. When $0.8 < E \leq 1.2$, the level of urbanization synchronizes with health production efficiency. When $E > 1.2$, the level of urbanization surpasses health production efficiency.

**2.1.4 Panel Tobit model.** The coupled and coordinated development of health production efficiency and new urbanization is influenced by various factors. Drawing on relevant

research findings and combining them with the actual situation of China, this paper selects the coupling coordination degree as the dependent variable and conducts econometric analysis on the explanatory variables, including population density, economic development level, government financial investment, urban-rural income gap, government health input level, individual health input level and health insurance level (Table 2). Since the coupling coordination degree lies between 0 and 1, ordinary least squares regression suffers from bias and inconsistency. Therefore, this paper uses the panel Tobit model to analyze the factors influencing the coupled and coordinated development of health production efficiency and new urbanization. In order to minimize the effect of multicollinearity and heteroskedasticity among variables on the model equation, all independent variables are logarithmized in this paper. The model is constructed as follows:

$$\begin{cases} Y_{it}^* = \beta X_{it} + \varepsilon_{it} \\ Y_{it} = Y_{it}^*, Y_{it}^* > 0 \\ Y_{it} = 0, Y_{it}^* \leq 0 \end{cases} \tag{5}$$

Where i is the evaluated unit, t is the years, $Y_{it}$ represents the dependent variable, $Y_{it}^*$ represents the latent variable, $X_{it}$ is the independent variable, $\beta$ is the estimated parameters, $\varepsilon_{it}$ is the disturbance term, and $\varepsilon_{it \sim N}(0,\sigma^2)$. Since the residuals can be viewed as observations of the error, we examined the normality of the residuals in IBM SPSS Statistics and found that they conformed to a normal distribution (Fig 1), and we assumed $\varepsilon_{it}$ is normally distributed under zero mean and constant variance [31,32].

## 2.2 Data sources

Due to data availability issues, this study selects 31 provinces in China (excluding Hong Kong, Macau, and Taiwan) as the research sample. Considering the large amount of missing data before 2003 and the impact of the COVID-19 pandemic on the research, this paper selected 2003–2018 as the study period. Data on indicators related to new urbanization were sourced from the annual China Statistical Yearbook, China Urban Statistical Yearbook, and local government statistical bulletins. The data on health inputs and outputs were obtained from the China Health and Health Statistics Yearbook, in which the health data for 2003 were obtained by interpolation due to missing data. With regard to the indicators of influencing factors, the raw data were obtained from the China Statistical Yearbook, and some of the data were obtained by calculation, such as the urban-rural income gap and individual health input level.

**Table 2. Influencing factors of coupling coordination.**

| Variable | Variable symbol | Variable description | Unit |
|---|---|---|---|
| Dependent variable | | | |
| Coupling coordination degree | D | Coupling coordination degree model calculation results | — |
| Independent variables | | | |
| Population density | pop | Population density | Person/km |
| Economic development level | GDP | Per capita GDP | Dollar |
| Government financial investment | gov | The proportion of regional financial expenditure in GDP | % |
| Urban-rural income gap | urg | Urban-rural per capita disposable income ratio | % |
| Government health input level | ing | The proportion of health expenditure in fiscal expenditure | % |
| Individual health input level | inp | The proportion of health care expenditure in consumption expenditure | % |
| Health insurance level | ins | Medical insurance participation rate | % |

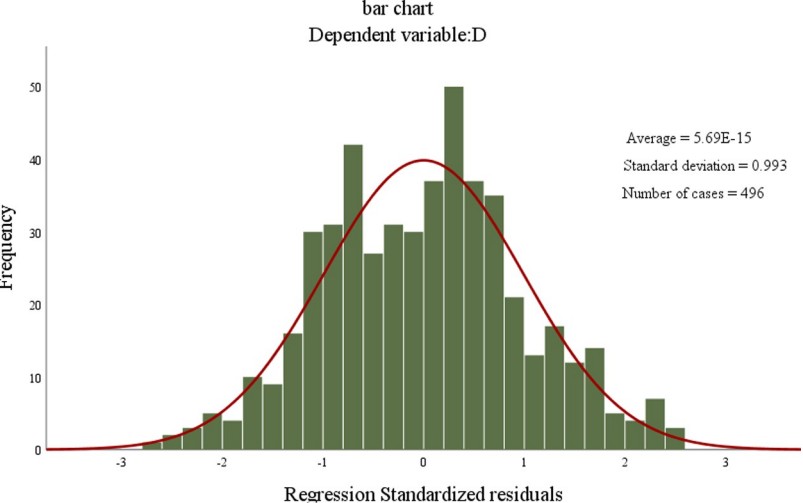

**Fig 1. Histogram of standardized residuals.**

## 3 Results

### 3.1 New urbanization level gradually improved in 2003–2018

During the study period, the mean value of the new urbanization in China increased steadily from 0.164 to 0.464 (Fig 2), indicating that the overall new urbanization level has been improved with rapid population movement, economic development, continuous improvement of public services and infrastructures, and gradual narrowing of the gap between urban and rural areas. At the regional level, The average level of new urbanization in the eastern, central, and western regions is 0.398, 0.269, and 0.238, respectively. The east region benefits from significant agglomeration effects in terms of industry, capital, and talent, resulting in a significantly higher level of new urbanization than the national average. The central and western new urbanization levels have gradually increased but still need to catch up to the national average. Therefore, to realize the coordinated development of the nation's economy, improving urbanization in the central and western regions is crucial. Examination of spatial patterns revealed that new urbanization levels in different provinces differed significantly from 2003 to 2018 (Fig 3). The new urbanization levels of Beijing, Jiangsu, Zhejiang, Shandong, and Guangdong have been significantly higher than other regions, while Tibet, Gansu, Guizhou, and Qinghai regions were at the lowest levels. In 2018, new urbanization levels were greater than 0.7 in Jiangsu and Shandong provinces, and according to Northam's three-stage urbanization theory, these two provinces have entered the later stable stage of urbanization. The new urbanization level in 28 provinces, excluding Tibet, remained between 0.3 and 0.7 in 2018. these regions have entered the accelerating development stage of urbanization.

### 3.2 Health production efficiency has an upward trend

From a temporal perspective (Fig 4), from 2003 to 2018, China's overall health production efficiency exhibited a fluctuating upward trend, rising from 0.87 in 2003 to 0.91 in 2018. The fluctuation pattern of national health production efficiency can be broadly divided into four stages. Firstly, 2003–2005 was a continuous decline stage, which can be attributed to the "SARS" epidemic outbreak in 2003. On the one hand, The increase in infectious disease cases significantly burdened the healthcare system, resulting in less optimistic health outputs. On

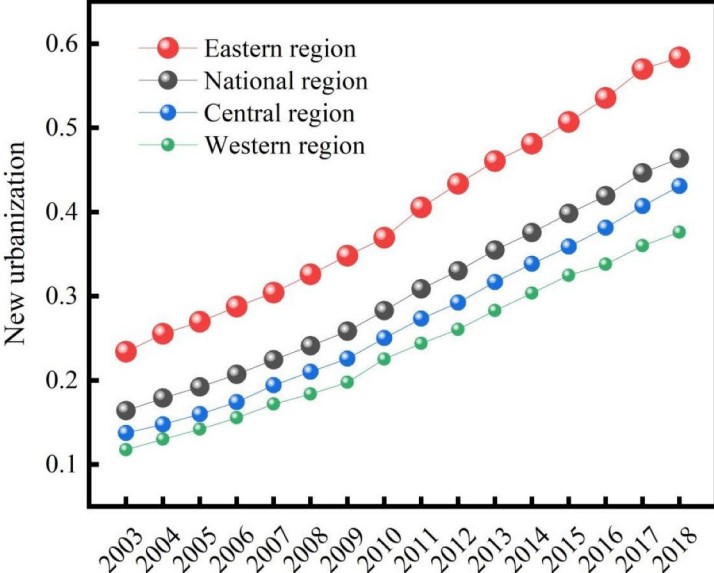

**Fig 2. Changing of new urbanization level in China in 2003–2018.**

the other hand, the nationwide impact of the "SARS" epidemic severely disrupted economic growth, making health inputs unable to meet the rapidly growing health demands. Secondly, the years 2006–2008 marked a stage of slow development. It was related to the government's emphasis on improving the level of medical services and accelerating the Construction of healthcare facilities in the post-epidemic era. Thirdly, the period of 2009–2011 witnessed a rapid decline. The adverse effects of the global financial crisis resulted in the closure of small and medium-sized enterprises and a consequential surge in unemployment. As a result, the level of health investment decreased for some impoverished individuals, resulting in a decrease in health production efficiency. Lastly, from 2012 to 2018, fluctuating growth was observed, with production efficiency hitting its lowest point in 2011 and subsequently fluctuating growth in the following years. After the new healthcare reform in 2009, the equity of healthcare resource allocation in China gradually improved [33–35]. The healthcare security system was gradually strengthened, and the capacity of medical services was further enhanced [36]. The differences in healthcare resources among regions gradually diminished [37], leading to a positive growth in health production efficiency.

In order to further explore the spatial and temporal evolution pattern of health production efficiency, the level of health productivity in 31 provinces was categorized into five levels using ArcGIS 10.2 software (Fig 5). From a regional perspective, health production efficiency shows regional agglomeration, with higher and highest levels mainly concentrated in the eastern regions, while the western and central regions are dominated by medium and lower levels of health production efficiency. From a provincial perspective, there are evident interprovincial disparities in health production efficiency in China, and these disparities are widening. It primarily manifests as an increase in provinces with low and high health production efficiency. During the study period, the health production efficiency in Shanghai, Anhui, Shandong, Jiangxi, Guizhou, and Tibet are all greater than 1, indicating a high level of health production efficiency. As seen in the graph of health inputs versus health outputs (Fig 6), Shanghai, Anhui, and Shandong belong to the high-input, high-output category due to their robust economic strength. The Jiangxi, Guizhou, and Tibet regions belong to the low-input, low-output

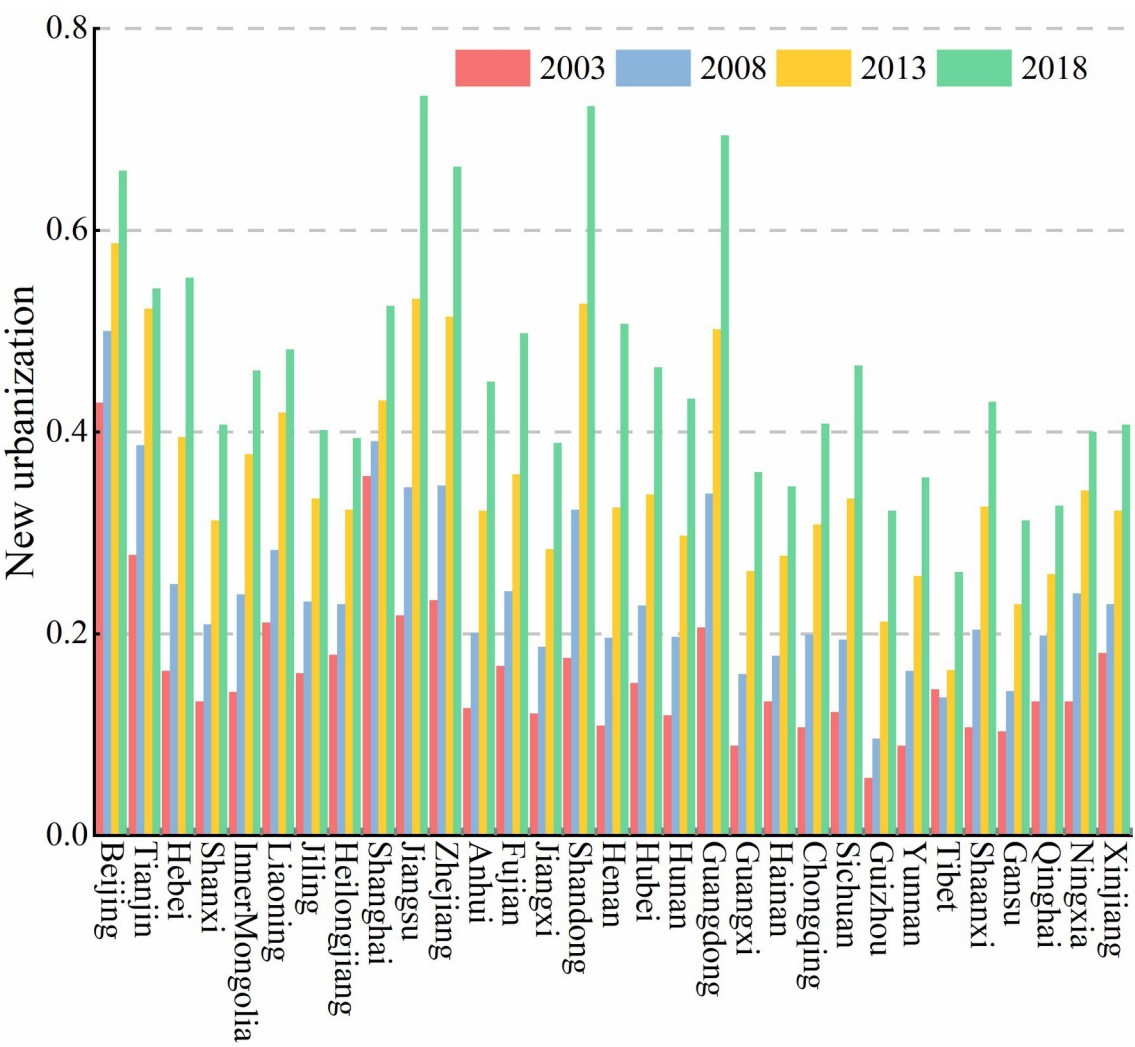

**Fig 3. New urbanization level of each region in China in 2003–2018.**

category. Although the level of economic development and health input in these places is significantly lower than the former, thanks to the fact that the residents of these regions have healthier habits and less social pressure, and are thus exposed to fewer health risks, resulting in the health needs can still be met with lower health inputs. The health production efficiency of Xinjiang, Qinghai, and Inner Mongolia was the lowest, consistently at low levels and lowest levels in each period. These regions belong to the high-input, low-output type, which may be related to the local climate and living habits.

## 3.3 The gradual narrowing of regional differences in health productivity

Using the Theil index, the spatial differentiation characteristics of health production efficiency in China were analyzed, as shown in Fig 7. From 2003 to 2018, the national Theil index for health production efficiency fluctuated between 0.072 and 0.286, exhibiting an overall trend of an initial increase followed by a decrease. This indicates that the spatial disparity in health production efficiency in China initially increased and later decreased over time. Among them, the Theil index was at its lowest in 2003, and this is because prior to the outbreak of the SARS

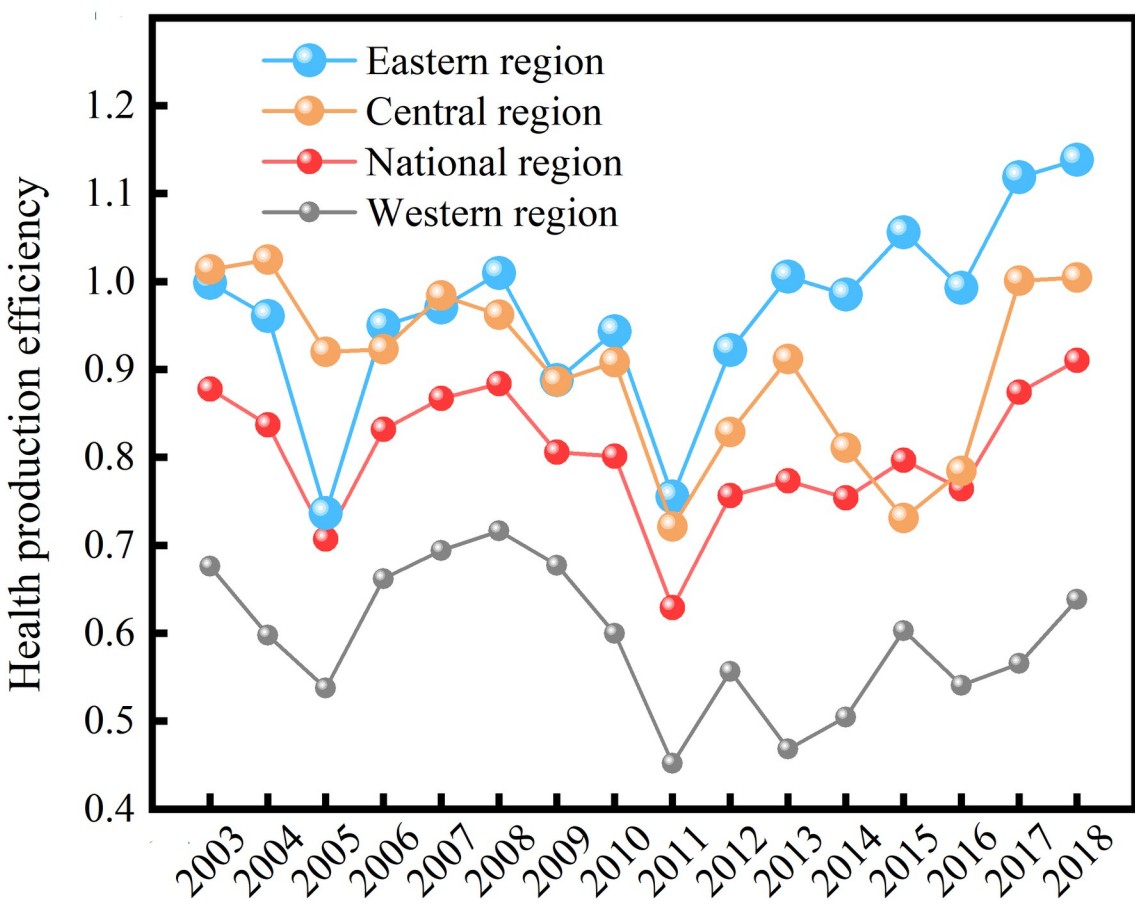

**Fig 4. The health production efficiency in China in 2003–2018.**

epidemic, there were relatively fewer spatial differences in medical resource allocation in China. The Theil index reached its peak in 2011 and gradually decreased to 0.075 in 2018. From 2003 to 2011, China learned the lessons of the SARS epidemic, and the government's awareness of health improvement increased. However, the regional disparities in this awareness and differences in health investment capacity led to an increase in regional disparities in health efficiency. Additionally, the differential growth rates of health demands in different regions also contributed to the expansion of spatial differences in health efficiency during this period. After 2011, with the maturation of the new healthcare reform policy and the introduction of the Healthy China strategy, the government's investment in health continuously increased, leading to a gradual reduction in regional disparities in health production efficiency. From a regional perspective, the Theil index showed a pattern of higher values in the western region, followed by the central and eastern regions, which aligns with the pattern of economic development disparities among Chinese cities. The decomposition of the Theil index results reveals that the proportion of the intra-region Theil index is significantly greater than that of the inter-region Theil index, indicating that intra-region differences mainly contribute to the regional disparities in health production efficiency in China. Therefore, achieving spatial balance in health production efficiency should focus on narrowing intra-region differences.

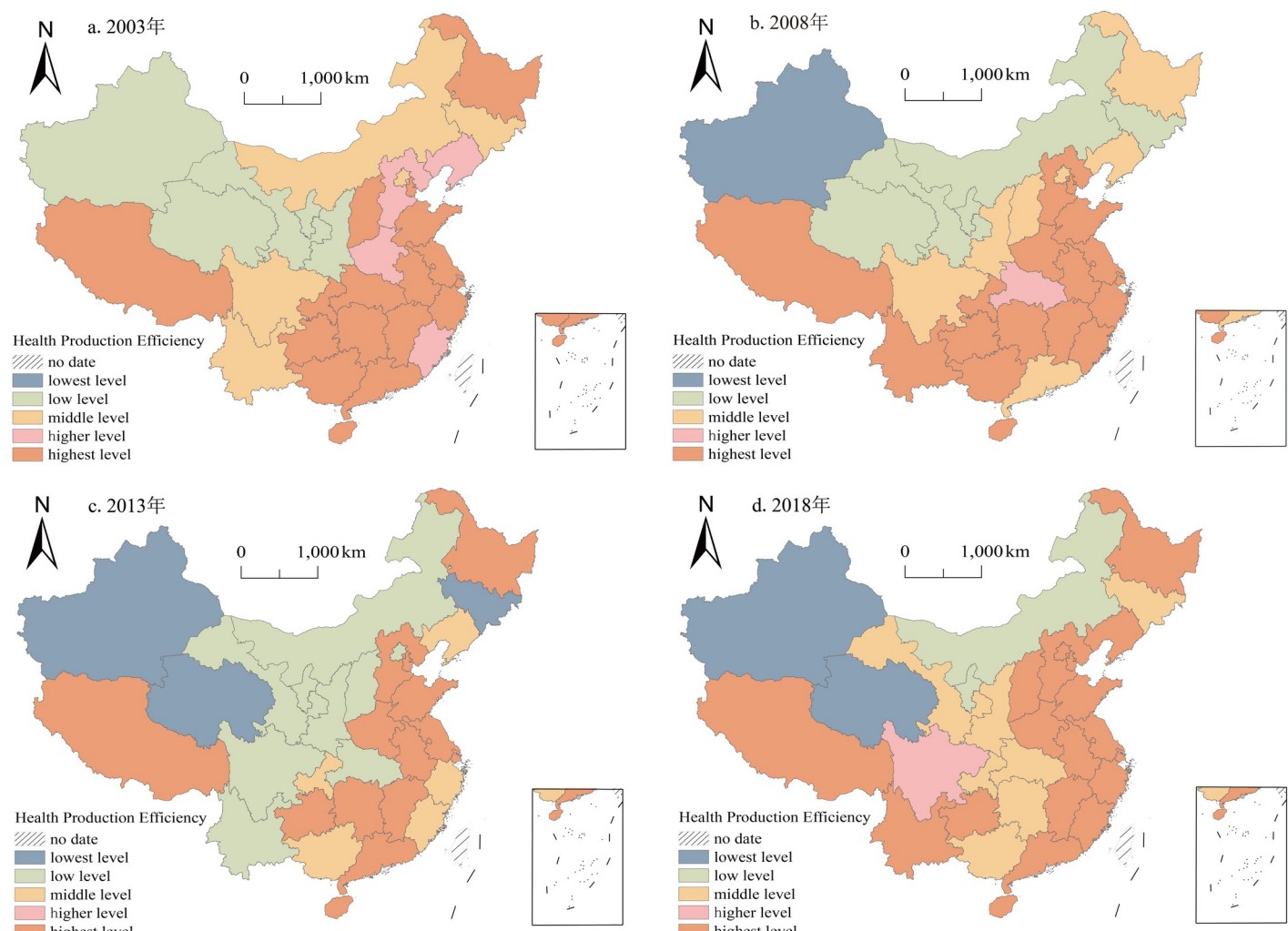

**Fig 5. The spatial evolution pattern of the health production efficiency in China in 2003–2018.** The basemap of China was downloaded from Natural Earth (located at: http://www.naturalearthdata.com/about/terms-of-use/), the figure is similar but not identical to the original image used in the study, and is therefore for illustrative purposes only. The data used was calculated by the author.

### 3.4 Spatio-temporal evolution analysis of the coupling coordination between health production efficiency and new urbanization

From 2003 to 2018, the coupling coordination degree between health production efficiency and new urbanization varied between 0.38 and 0.95 across different provinces. The coordination types transformed on the verge of incoordination to barely coordinated, primary, moderate, and eventually achieving good and high-quality coordination (Fig 8). Among them, cities on the verge of incoordination, barely coordinated, and primary coordination was reduced by 16%, 23%, and 29%, respectively. In contrast, cities with moderate coordination, good coordination, and high-quality coordination increased by 22%, 23%, and 23%, respectively. It indicates that with the progress of new urbanization and the implementation of the Healthy China strategy, the two systems are evolving towards a more positive and orderly direction, with the coupling coordination situation continuously improving.

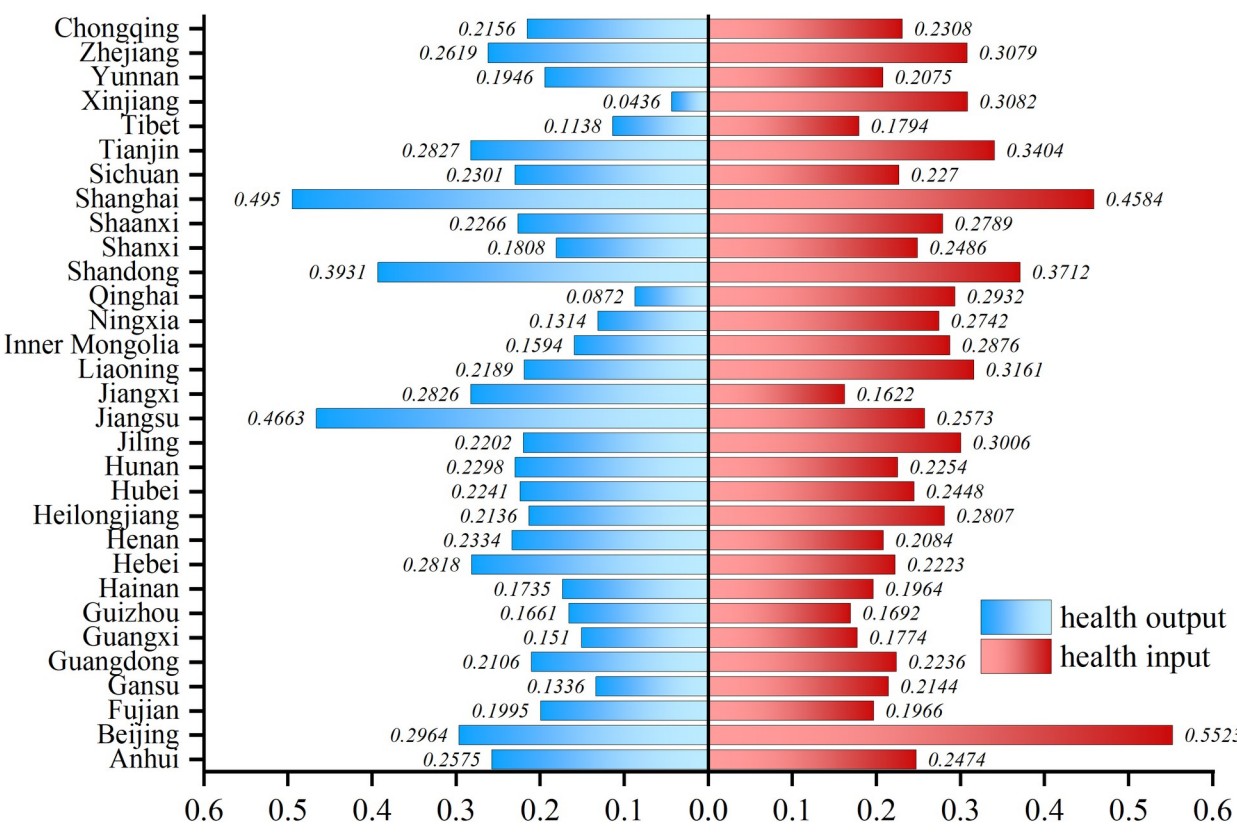

**Fig 6. Health inputs versus health outputs.**

Looking at the relative development status of health production efficiency and new urbanization (Fig 9), it has experienced new urbanization lagging behind health production efficiency, new urbanization synchronizing with health production efficiency, and new urbanization surpassed health production efficiency (referred to as lagging, synchronous, and

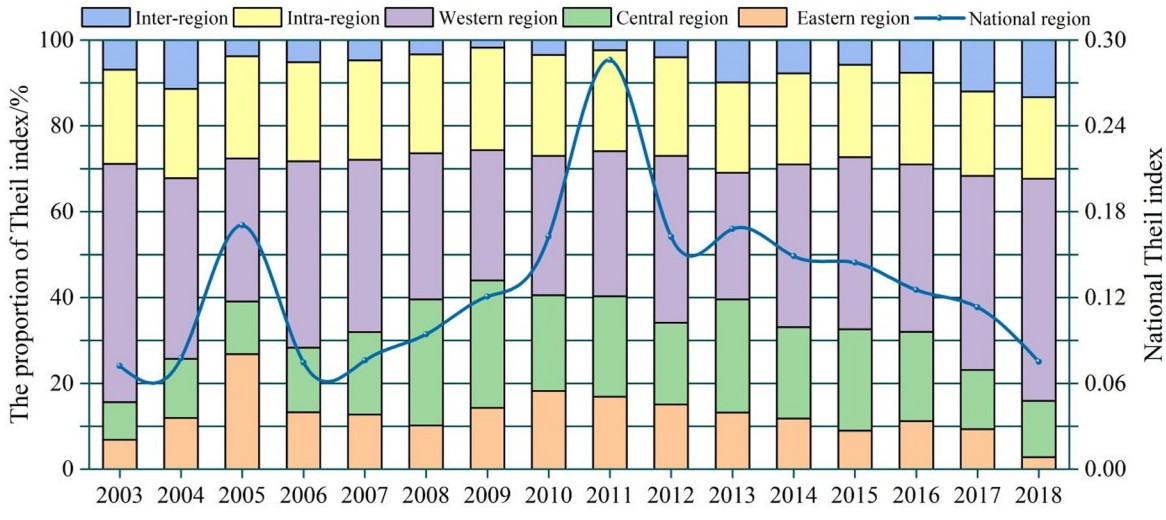

**Fig 7. Theil index decomposition of the health production efficiency in China in 2003–2018.**

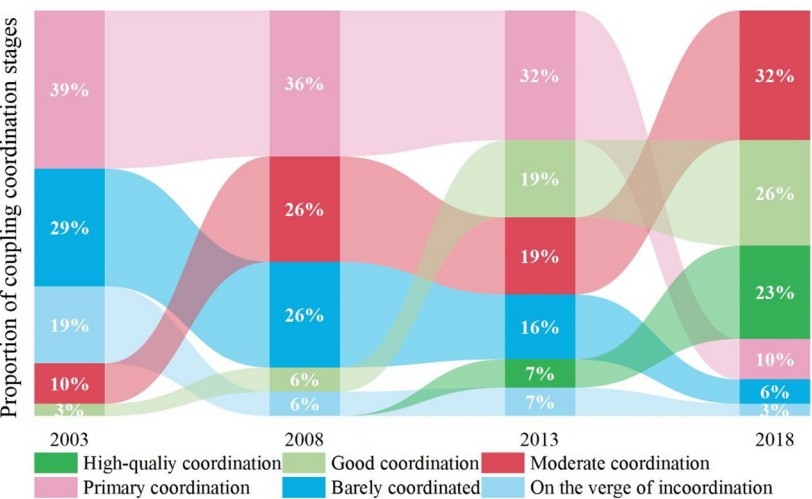

**Fig 8. Proportion of coupling coordination stage between the health production efficiency and new urbanization.**

advanced states). It can be seen that during the study period, the number of regions in the lagging stage gradually decreased, while the number of areas in the advanced and synchronous stages gradually increased. It can be attributed to China's "Great Leap Forward" style of urbanization, which resulted in health production efficiency needing to improve to keep up with the rapid pace of new urbanization.

This study visualizes the classified results of coupling coordination to further explore the interactive relationship between health production efficiency and new urbanization. It selects four periods, namely 2003, 2008, 2013, and 2018, for demonstration (Fig 10). From the perspective of changes in coupling coordination types, Qinghai Province has consistently remained on the verge of the incoordination stage from 2003 to 2018. Eight regions, namely Ningxia, Xinjiang, Inner Mongolia, Guangxi, Hubei, Hainan, Tibet, and Shanghai, have all seen an upgrade in their coupling coordination types. Among them, Ningxia and Xinjiang have transitioned from the verge of incoordination stage to the barely coordinated stage; Inner

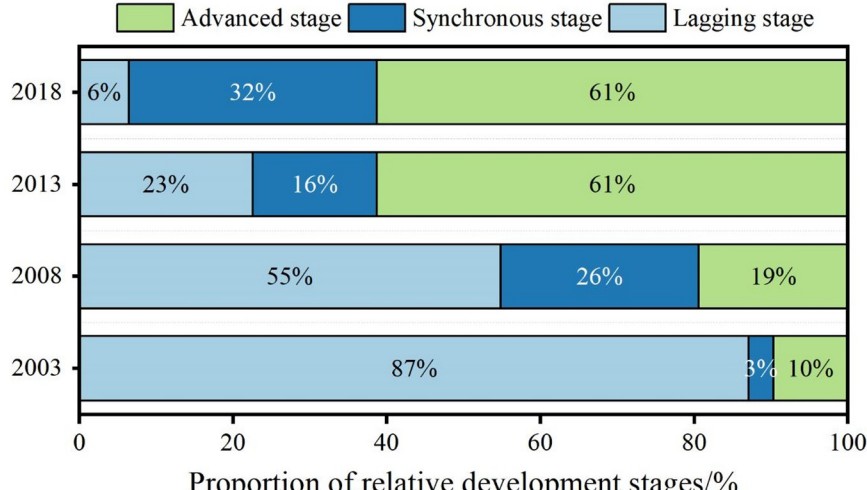

**Fig 9. Proportionof relative development stage between the health production efficiency and new urbanization.**

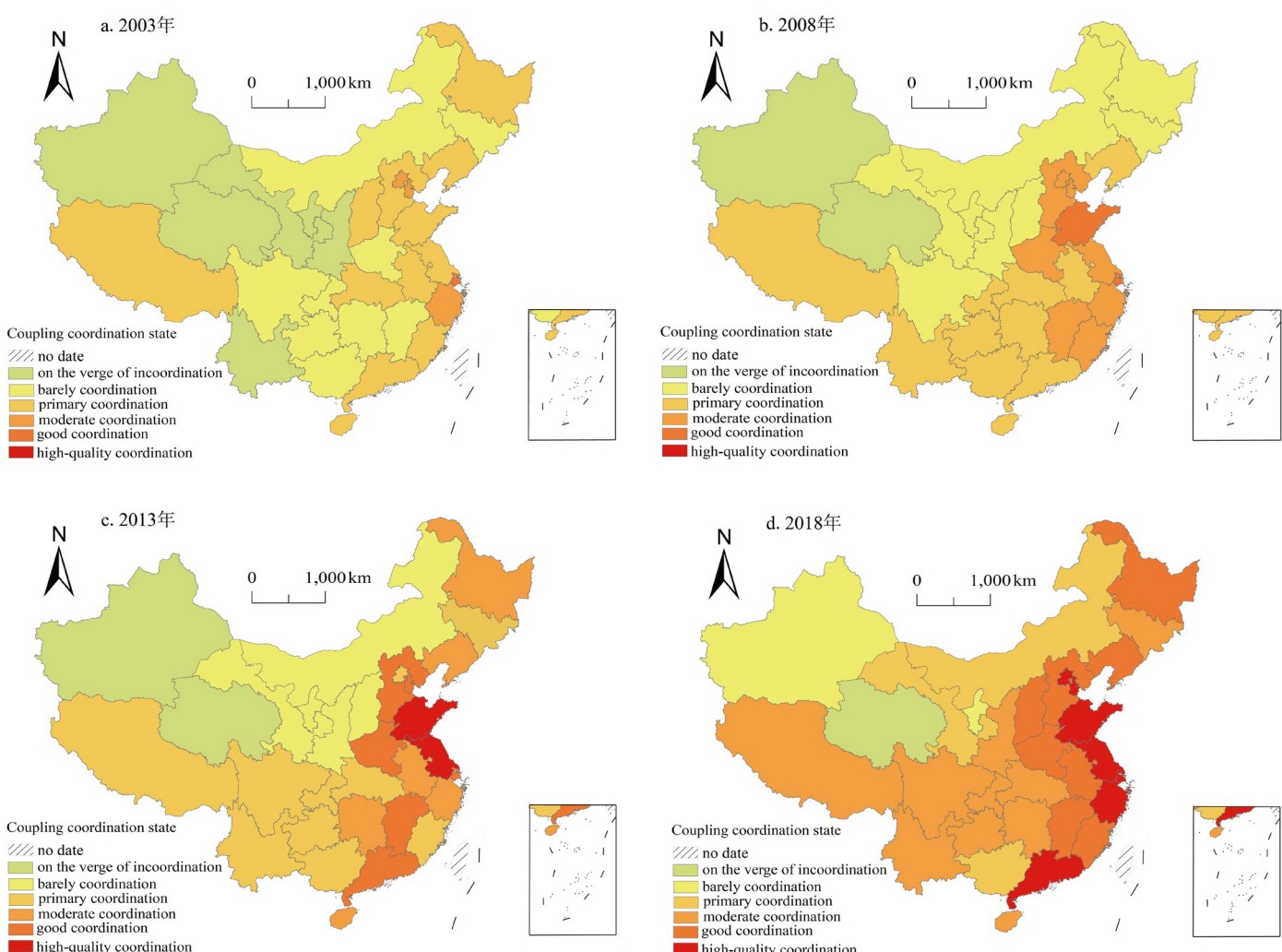

**Fig 10. The spatial evolution pattern of coupling coordination degree of the health production efficiency and new urbanization in China in 2003–2018.** The basemap of China was downloaded from Natural Earth(located at: http://www.naturalearthdata.com/about/terms-of-use/), the figure is similar but not identical to the original image used in the study, and is therefore for illustrative purposes only. The data used was calculated by the author.

Mongolia and Guangxi have advanced from the barely coordinated stage to the primary coordination stage; Hubei, Hainan, and Tibet have progressed from the primary coordination stage to the moderate coordination stage; and Shanghai has elevated from the good coordination stage to the high-quality coordination stage. Fifteen regions, including Gansu, Jilin, Hunan, Chongqing, Sichuan, Guizhou, Hebei, Shanxi, Liaoning, Heilongjiang, Anhui, Fujian, Beijing, Tianjin, and Zhejiang, have all experienced a two-level increase in their coupling coordination types. Gansu has transitioned from the verge of incoordination stage to the primary coordination stage; Jilin, Hunan, Chongqing, Sichuan, and Guizhou have advanced from the barely coordinated stage to the moderate coordination stage; Hebei, Shanxi, Liaoning, Heilongjiang, Anhui, and Fujian have progressed from the primary coordination stage to the good coordination stage; and Beijing, Tianjin, and Zhejiang have elevated from the moderate coordination stage to the high-quality coordination stage. Seven regions, namely Yunnan, Shaanxi, Jiangxi, Henan, Jiangsu, Shandong, and Guangdong, have all experienced a three-level increase in their coupling coordination types. Yunnan and Shaanxi have transitioned from the

verge of incoordination stage to the moderate coordination stage; Jiangxi and Henan have advanced from the barely coordinated stage to the good coordination stage; and Jiangsu, Shandong, and Guangdong have progressed from the primary coordination stage to the high-quality coordination stage. Overall, the coupling coordination between health production efficiency and new urbanization in China shows a general upward trend, with 71% of provinces experiencing a two-level or higher increase in their coupling coordination types, showing a significant leap in hierarchical transition. It suggests that the gap between health production efficiency and new urbanization level in China is gradually narrowing over time.

From a spatial perspective, the coupling and coordination degree between health production efficiency and new urbanization exhibit a pattern of "higher in the east and lower in the west, higher in the south and lower in the north." There are significant disparities among provinces, and spatial agglomeration phenomena are evident. In 2003, areas with high coupling coordination degrees were distributed patchy, with Beijing and Shanghai as the centers, while areas with low coupling coordination degrees were concentrated in the northwest region, aligning with China's economic development. By 2008, the regional disparities in coupling coordination degree among China's eastern, central, and western regions gradually narrowed. Only Qinghai and Xinjiang provinces were on the verge of incoordination. In contrast, provinces in the primary coordination stage continued to advance westward, forming a V-shaped spatial distribution from Tibet to Liaoning. The range of provinces in a stage of good coordination and moderate coordination gradually expanded and distributed along the coastline. By 2013, Jiangsu and Shandong took the lead in entering the stage of high-quality coordination. The proportion of provinces in the primary coordination stage or above increased from 64% in 2008 to 71%. By 2018, seven provinces, namely Jiangsu, Shandong, Beijing, Tianjin, Shanghai, Zhejiang, and Guangdong, entered the stage of high-quality coordination, accounting for 22.6% of the total sample. These regions benefit from superior geographical locations, open policy conditions, and high levels of industrial agglomeration. Their level of new urbanization is at the forefront nationwide, providing a solid economic foundation and a high-quality talent guarantee for the development of health efficiency. The two systems have formed strong synergistic development effects.

## 3.5 Analysis of factors influencing the coupling and coordination of the two systems

Based on the model constructed in the previous section, a Tobit regression analysis was conducted using Stata 17.0 software to examine the impact factors of the coupling and coordination of health production efficiency and new urbanization from 2003 to 2018. The results are presented in Table 3.

The regression analysis results indicate that the coefficient of population density is significantly positive at the 1% level. It suggests that increasing population density can enhance coupling coordination between the two systems. This is because the increase in population density can increase the stock of regional human capital and promote the clustering of scientific and innovative talents and technologies. In addition, the increase in population density will also lead to medical facilities and other health resources to cover a larger population, which is conducive to improving the efficiency of their utilization. Thereby facilitating the coordinated development of the two systems. From an economic standpoint, one standard deviation increase in population density is correlated with a rise of 0.07 coupling coordination degree. This amount corresponds to 10.45% of our sample's average coupling coordination degree. The coefficient of the economic development level is significantly positive at the 1% level, indicating that a more robust economy can provide both solid financial support and a material

**Table 3. Regression results of influencing factors of coupling coordination.**

| Explanatory variables | Regression coefficient | Standard error | Z value | P value |
|---|---|---|---|---|
| lnpop | 0.0498*** | 0.0095 | 5.23 | 0.000 |
| lngdp | 0.0508*** | 0.0129 | 3.95 | 0.000 |
| lngov | 0.0462** | 0.0218 | 2.12 | 0.034 |
| lnurg | -0.0344 | 0.0404 | -0.85 | 0.395 |
| lning | 0.0732*** | 0.0196 | 3.74 | 0.000 |
| lninp | -0.0667** | 0.0273 | -2.44 | 0.015 |
| lnins | -0.0077 | 0.0054 | -1.42 | 0.155 |

Note: ***, **, * indicate significance at the 1 percent, 5 percent, and 10 percent levels, respectively.

basis for new urbanization, as well as promote health inputs, improve health production efficiency, and contribute to the coordinated development of the two systems. Therefore, economic development should continue to be steadily promoted in the future. Specifically, one standard deviation increase in per capita GDP is correlated with a rise of 0.04 coupling coordination degree. This amount corresponds to 5.97% of our sample's average coupling coordination degree. The coefficient of government financial investment is significantly positive at the 5% level, indicating that macro policy guidance by the government has a positive effect on the coordinated development of the two systems. From an economic standpoint, one standard deviation increase in government financial investment is correlated with a rise of 0.02 coupling coordination degree. This amount corresponds to 2.99% of our sample's average coupling coordination degree. The more the government spends, the greater its ability to improve public service facilities, the environment and ecology, and people's livelihoods, thus contributing to the enhancement of new urbanization and health promotion. Therefore, the macro-control role of the government should be brought into full play, and its capacity should be further strengthened to promote the coordinated development of the two systems through policy favoritism and financial support. The coefficient of the government health input level is significantly positive at the 1% level, indicating that the government health input level has a positive effect on the coordinated development of the two systems. From an economic standpoint, one standard deviation increase in government health input level is correlated with a rise of 0.02 coupling coordination degree. This amount corresponds to 2.99% of our sample's average coupling coordination degree. During the study period, China's new urbanization level has increased dramatically and is in the stage of rapid development. The lagging health production efficiency has become the main factor restricting the coordinated development of new urbanization and health production efficiency. Government health investment is conducive to reducing the burden of residents' visits to the doctor and promoting the growth of health productivity, which in turn is conducive to the coordinated development of the two systems. The coefficient of the individual health input level is significantly negative at the 5% level, indicating that the individual health investment level has an inhibitory effect on the coordinated development of the two systems. From an economic standpoint, one standard deviation increase in individual health input level is correlated with a decrease of 0.02 coupling coordination degree. This amount corresponds to 2.99% of our sample's average coupling coordination degree. It may be because a higher proportion of individual health expenses increases the likelihood of poverty, resulting in less time and funds being allocated to production, life, and other aspects, thus reducing opportunities to create value for the economy and society, which is not conducive to promoting the coordinated development of health and the economy. There is no significant correlation between the urban-rural income gap and the health

insurance level and the coupling coordination of health production efficiency and new urbanization.

## 4 Discussion

The coordinated development of health production efficiency and new urbanization is conducive to promoting residents' well-being and socioeconomic development. The coordinated development of the two systems has two meanings. The first is that the development of new urbanization must not undermine health productivity. At the same time, health productivity must not inhibit the development of new urbanization, which is the most basic. The second level is that the two systems reinforce each other. If the urban habitat is destroyed to increase the urbanization level, the health of the corresponding regional population health will deteriorate, which will, in turn, inhibit the development of urbanization. In addition, urbanization can provide a material basis for health productivity growth, and it is also unrealistic to pursue health production efficiency away from urbanization. In the policy-making to promote the coordinated development of the two systems, the development differences among cities and the actual health problems faced by each province should be considered. Cities with a high new urbanization level, such as Beijing, Jiangsu, Zhejiang, and Shandong, already have a solid economic foundation; these cities should pay more attention to improving health productivity and sacrifice some economic benefits when necessary. Cities with a low new urbanization level, such as Tibet, Xinjiang, and Gansu, need a more robust economic foundation. Pursuing rapid economic growth in the short term is their primary goal. The requirements can be appropriately reduced to avoid hindering economic growth. Population density, the level of economic development, and government financial investment have a positive impact on the coordinated development of the two systems, which means that policymakers, while rationally introducing talents and technologies and continuously promoting economic development, should also better play the role of the government's macro-control and improve the efficiency of the allocation of resource factors, to push forward the coordinated development of the efficiency of new urbanization and health productivity efficiency.

Like most studies, this one has some shortcomings. Firstly, In this paper, the health production decision unit is narrowly defined as the health care system. Therefore, it overlooks other social and environmental variables related to health input, such as health education, health environment, and physical exercise. Future research should comprehensively consider the impact of these factors and deeply explore the reasons affecting the level of health production efficiency and its regional differences. Secondly, limited to the availability of health output variables, the spatial scale chosen in this study is at the provincial level, which is only suitable for macro strategic reference. Finally, the selection of evaluation indicators for new urbanization and explanatory variables for coupling coordination may need to be more comprehensive. In future research, these aspects will be continuously supplemented to deeply study the two systems' interactive relationship and driving factors.

## 5 Conclusions

This paper takes 31 provinces in China as the research object. Firstly, the entropy value method and the super-efficient SBM model were used to measure the development level of China's new urbanization and health production efficiency, respectively. Then, the Theil index was employed to measure the spatial disparities in health production efficiency. By introducing the coupling coordination model and the relative development degree model, the evolution characteristics of the coupling coordination relationship between the new urbanization and health production efficiency in China were revealed from two dimensions: temporal evolution

and spatial pattern. Finally, the Tobit regression model was used to investigate the influencing factors of coupling coordination between the two systems. The research findings are as follows:

1. From the perspective of comprehensive development level, the level of new urbanization and health production efficiency in China has shown a fluctuating upward trend during the research period. However, the development level is still relatively low, leaving ample room for improvement. At the regional level, both exhibit a distribution pattern of decreasing levels from the eastern to central to western regions, demonstrating clear spatial consistency.

2. In terms of spatial inequality of health production efficiency, the year 2011 serves as a critical point of the change in health productivity Theil index, which showed a fluctuating upward trend in the earlier period, reaching its peak in 2011, and gradually declined from 2012 to 2018, indicating that regional differences in health production efficiency in China initially widened and then narrowed. The western region exhibits the greatest spatial disparity in health production efficiency at the regional level, followed by the central region, while the eastern region has the smallest disparity. Moreover, intra-regional differences are the primary source of spatial differences in health production efficiency in China.

3. From the spatiotemporal evolution of coupling coordination, China's health production efficiency and new urbanization have shown a positive developmental trend during the research period, which increased by 32.16% from 0.597 in 2013 to 0.789 in 2018. Regarding the evolution of coupling coordination types, from 2003 to 2018, 3.23% of provinces maintained the same level of coupling coordination, while 96.77% of provinces moved up to a higher level. Among them, 50% of provinces moved up two levels in terms of coupling coordination. At the spatial level, the coupling coordination between health production efficiency and new urbanization in China exhibits a pattern of "higher in the east and lower in the west, higher in the south and lower in the north," with significant interprovincial differences and strong spatial clustering.

4. In terms of the factors influencing the coupling coordination between health production efficiency and new urbanization, population density, economic development level, government fiscal investment, and government health investment have a positive impact on the coupling coordination of the two systems. In contrast, individual health investment has a negative effect.

## Supporting information

**S1 File. Health production efficiency indicators.**
(XLSX)

**S2 File. Influencing indicators.**
(XLSX)

**S3 File. New urbanization indicators.**
(XLSX)

## Author Contributions

**Conceptualization:** Fang Zhang.

**Data curation:** Fang Zhang, Yuhan Du.

**Formal analysis:** Haili Zhao, Fang Zhang.

**Investigation:** Yuhan Du, Jialiang Li.

**Methodology:** Fang Zhang, Minghui Wu.

**Supervision:** Haili Zhao, Fang Zhang.

**Visualization:** Fang Zhang.

**Writing – original draft:** Fang Zhang.

**Writing – review & editing:** Haili Zhao.

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
