## [Decision Letter · Decision Letter 0]

11 Oct 2023

PONE-D-23-24900The coupling coordination characteristics of China’s health production efficiency and new urbanization and its influencing factorsPLOS ONE

Dear Dr. Zhang,

Thank you for submitting your manuscript to PLOS ONE. After careful consideration, we feel that it has merit but does not fully meet PLOS ONE’s publication criteria as it currently stands. Therefore, we invite you to submit a revised version of the manuscript that addresses the points raised during the review process.

We look forward to receiving your revised manuscript.

Kind regards,

Yu Zhou

Academic Editor

PLOS ONE

4. We note that Figures 3 and 7 in your submission contain [map/satellite] images which may be copyrighted. All PLOS content is published under the Creative Commons Attribution License (CC BY 4.0), which means that the manuscript, images, and Supporting Information files will be freely available online, and any third party is permitted to access, download, copy, distribute, and use these materials in any way, even commercially, with proper attribution. For these reasons, we cannot publish previously copyrighted maps or satellite images created using proprietary data, such as Google software (Google Maps, Street View, and Earth). For more information, see our copyright guidelines: http://journals.plos.org/plosone/s/licenses-and-copyright.

a. You may seek permission from the original copyright holder of Figures 3 and 7 to publish the content specifically under the CC BY 4.0 license. 

5. Please remove your figures from within your manuscript file, leaving only the individual TIFF/EPS image files, uploaded separately. These will be automatically included in the reviewers’ PDF.

Reviewers' comments:

Reviewer's Responses to Questions

**Comments to the Author**

1. Is the manuscript technically sound, and do the data support the conclusions?

Reviewer #1: Yes

Reviewer #2: Yes

2. Has the statistical analysis been performed appropriately and rigorously? 

Reviewer #1: Yes

Reviewer #2: Yes

3. Have the authors made all data underlying the findings in their manuscript fully available?

Reviewer #1: Yes

Reviewer #2: Yes

4. Is the manuscript presented in an intelligible fashion and written in standard English?

Reviewer #1: No

Reviewer #2: Yes

5. Review Comments to the Author

Reviewer #1: In general, I think this article is not well organized.

1. The abstract is too long and the point is not obvious;

2. The structure of many paragraphs in the introduction is also unclear, for example, the First of all is not followed by a subsequent expression. Many references are also irregular. For example, Yu Jiali conducted a full-factor analysis…;

3. The section of Materials and Methods is too redundant. It is proposed to merge 2.1.1 with 2.1.3; to merge 2.1.2 with 2.1.4; to merge 2.1.6 with 2.1.7. In addition, The Thiel index is a very common indicator and does not need to be included in the methodology.

4. It is recommended that Figure 2 not be presented with a radar chart, but with a line chart similar to Figure 1, so that it is easier to compare the trends between the two.

5. Why is only the spatial evolution pattern of the health production efficiency introduced? Why was the spatial evolution pattern of the new urbanization level not considered?

6. The analysis of influencing factors is too simple, and some explanations are far-fetched. For example, why is coefficient of population density significantly positive? Your explanation lacks conviction.

7. Some expressions in the discussion section do not correspond to reality. For example, what you said “For the first time, this article examines the interrelationship…”

Reviewer #2: I have reviewed the manuscript titled "The coupling coordination characteristics of China’s health production efficiency and new urbanization and its influencing factors." While the study addresses an important aspect of the relationship between health production efficiency and urbanization in China, there are several comments and concerns regarding the manuscript:

Lack of Clear Research Objectives:

The article mentions measuring health production efficiency and new urbanization in Chinese provinces, but it does not clearly state the research objectives or specific research questions. The manuscript should provide a more concise and focused statement of its goals.

Methodological Issues:

The article mentions the use of the super-efficient SBM model, entropy value method, Thiel index, coupling coordination degree model, and Tobit regression analysis, but it lacks details on the methodology employed. A more comprehensive explanation of these methods, data sources, and their application is necessary to understand the study's validity and replicability.

Normality of error term:

I highly recommend empirically evaluating the normality of the given error term in equation 5, and its results in the revised paper. Alternatively, the author can add the given assumption with the addition of given studies in the revised article as “The given error term in equation 5 is assumed to be normally distributed at zero mean value and constant variance [1,2].

[1] Understanding farmers’ intention and willingness to install renewable energy technology: A solution to reduce the environmental emissions of agriculture.

[2] Handling endogenous regressors using copulas: A generalization to linear panel models with fixed effects and correlated regressors

Limited Presentation of Findings:

The article provides general statements about the trends in health production efficiency and new urbanization but lacks specific findings or quantitative results. Readers need more concrete data and insights to assess the significance of these trends.

Explanation of Coupling Coordination:

While the article discusses the coupling coordination degree between health production efficiency and new urbanization, it does not clarify what this means or its significance for residents' well-being and socioeconomic development. A more detailed explanation of this concept and its practical implications is needed.

Limited Discussion of Influencing Factors:

The article mentions factors such as population density, economic development, government financial investment, and government health investment affecting coupling coordination. However, it does not elaborate on how these factors impact the relationship or their policy implications.

Clarity and Organization:

The article would benefit from improved clarity and organization. The flow of information is somewhat fragmented, making it challenging for readers to follow the logical progression of the study.

6. PLOS authors have the option to publish the peer review history of their article (what does this mean?). If published, this will include your full peer review and any attached files.

Reviewer #1: No

Reviewer #2: No

---

## [Author Response · Author response to Decision Letter 0]

13 Jan 2024

Dear reviewers

Thank you for your letter. We were pleased to know that our work was rated as potentially acceptable for publication in the Journal, subject to adequate revision. We thank the academic editor and reviewers for the time and effort that they have put into reviewing the previous version of the manuscript. We have revised the manuscript and image format according to the journal's requirements. The comments of the academic editor and reviewers are particularly important for the improvement of the content and quality of this study. We have made extensive changes to the article as requested by the reviewers. We have highlighted your requested changes with a yellow background and attached a revised version for your convenience. Some of the changes may be repeated, we have prioritised highlighting them in yellow to make them easier for you to read. We are very grateful to you and the other one reviewers for your comments on our paper. We look forward to your response. Thank you and best wishes.

Response to Reviewer #1

Reviewer #1: In general, I think this article is not well organized.

1. The abstract is too long and the point is not obvious;

Modification Description: We are very grateful for the reviewer's comments, and we have revised it according to the revision requirements. For the problem that the abstract is too long and unfocused. We have condensed the presentation of the abstract, focusing on highlighting the significance, content, corresponding research methodology, and important research conclusions of the study. 

2. The structure of many paragraphs in the introduction is also unclear, for example, the First of all is not followed by a subsequent expression. Many references are also irregular. For example, Yu Jiali conducted a full-factor analysis…;

Modification Description: We are very grateful for the reviewer's comments, and we have revised it according to the revision requirements. We have sorted out the structure and logic of the paragraphs in the introductory section, The first paragraph introduces the interaction between health production efficiency and new urbanization, the second and third paragraphs are a review of the research on new urbanization and health production efficiency, respectively, and the last paragraph summarizes the research questions and research objectives of this paper. In addition, we have standardized the language expression and cited the references in a standardized way. 

3.The section of Materials and Methods is too redundant. It is proposed to merge 2.1.1 with 2.1.3; to merge 2.1.2 with 2.1.4; to merge 2.1.6 with 2.1.7. In addition, The Thiel index is a very common indicator and does not need to be included in the methodology.

Modification Description: We are very grateful for the reviewer's comments, and we have revised it according to the revision requirements. We have merged 2.1.1 with 2.1.3, merged 2.1.2 with 2.1.4, and 2.1.6 with 2.1.7. and the explanation of the common method of the Terrell Index has been removed, which makes the Materials and Methods section more concise and clearer.

4. It is recommended that Figure 2 not be presented with a radar chart, but with a line chart similar to Figure 1, so that it is easier to compare the trends between the two.

Modification Description: We are very grateful for the reviewer's comments, and we have revised it according to the revision requirements. Figure 2 (now Figure 4) has been changed to be presented in a line graph, which is more conducive to comparing the development trends of the two systems. The new urbanization level shows a steady upward trend, while the level of health production efficiency fluctuates upward. 

5. Why is only the spatial evolution pattern of the health production efficiency introduced? Why was the spatial evolution pattern of the new urbanization level not considered?

Modification Description: We are very grateful for the reviewer's comments, and we have revised it according to the revision requirements. A bar chart reacting to the spatial and temporal changes of new urbanization in each province from 2003 to 2018 has been added to the revised draft (Fig. 2), which can visually reflect the inter-provincial development trend and development differences of new urbanization. Compared with the health production efficiency, the change of new urbanization is more stable, showing a year-by-year upward trend. The regional level exhibits a decreasing way from the eastern to central to western regions. The inter-provincial differences are obvious, and most of the cities entered the rapid development stage in 2018. 

6. The analysis of influencing factors is too simple, and some explanations are far-fetched. For example, why is coefficient of population density significantly positive? Your explanation lacks conviction.

Modification Description: We are very grateful for the reviewer's comments, and we have revised it according to the revision requirements. We have explained in more detail the influences of population density, economic development, government financial inputs, and government health inputs. 

7. Some expressions in the discussion section do not correspond to reality. For example, what you said “For the first time, this article examines the interrelationship…”

Modification Description: We are very grateful for the reviewer's comments, and we have revised it according to the revision requirements. We have deleted descriptions in the discussion that do not correspond to reality, added the concept and practical implications of the coordinated development of new urbanization and health production efficiency, and put forward recommendations related to the coordinated development of the two systems. 

Response to Reviewer #2

Reviewer #2: I have reviewed the manuscript titled "The coupling coordination characteristics of China’s health production efficiency and new urbanization and its influencing factors." While the study addresses an important aspect of the relationship between health production efficiency and urban.

1.Lack of Clear Research Objectives:

The article mentions measuring health production efficiency and new urbanization in Chinese provinces, but it does not clearly state the research objectives or specific research questions. The manuscript should provide a more concise and focused statement of its goals.

Modification Description: We are very grateful for the reviewer's comments, and we have revised it according to the revision requirements. We have clearly stated the purpose and significance of the research in this paper in the Abstract and Introduction, and focus on summarizing the specific research questions of this paper in the Introduction. 

2.Methodological Issues:

The article mentions the use of the super-efficient SBM model, entropy value method, Thiel index, coupling coordination degree model, and Tobit regression analysis, but it lacks details on the methodology employed. A more comprehensive explanation of these methods, data sources, and their application is necessary to understand the study's validity and replicability.

Modification Description: We are very grateful for the reviewer's comments, and we have revised it according to the revision requirements. We have added the formula of the entropy value method and improved the source of data used. We supplemented the software for the calculation of methods such as the super-efficiency SBM model, entropy method and coupling coordination model. Among them, the entropy value method was carried out in Stata17.0, and the super-efficiency SBM model was carried out in the software MaxDEA. The calculation of the coupling coordination degree was realized in Excel, and its spatial visualization was carried out in ArcMap 10.2. 

3.Normality of error term: 

I highly recommend empirically evaluating the normality of the given error term in equation 5, and its results in the revised paper. Alternatively, the author can add the given assumption with the addition of given studies in the revised article as “The given error term in equation 5 is assumed to be normally distributed at zero mean value and constant variance [1,2].

[1] Understanding farmers’ intention and willingness to install renewable energy technology: A solution to reduce the environmental emissions of agriculture.

[2] Handling endogenous regressors using copulas: A generalization to linear panel models with fixed effects and correlated regressors.

Modification Description: We are very grateful for the reviewer's comments, and we have revised it according to the revision requirements. We have tested the normality of the error term in equation 5. Since the residuals can be viewed as observations of the error, we tested the normality of the residuals in IBM SPSS Statistics and found that they conformed to a normal distribution (Fig. 1). With reference to the relevant literature we assumed that the error term given in equation 5 is normally distributed under zero mean and constant variance. 

4.Limited Presentation of Findings:

The article provides general statements about the trends in health production efficiency and new urbanization but lacks specific findings or quantitative results. Readers need more concrete data and insights to assess the significance of these trends.

Modification Description: We are very grateful for the reviewer's comments, and we have revised it according to the revision requirements. We have enriched the quantitative analysis of new urbanization and health production efficiency. We added a bar chart of the temporal and spatial changes of new urbanization in each province from 2003-2018 to analyze their interprovincial development characteristics and stages. In the section on health production efficiency, a graph comparing health inputs and health outputs has been added as an important basis for analyzing changes in health productivity. 

5.Explanation of Coupling Coordination:

While the article discusses the coupling coordination degree between health production efficiency and new urbanization, it does not clarify what this means or its significance for residents' well-being and socioeconomic development. A more detailed explanation of this concept and its practical implications is needed.

Modification Description: We are very grateful for the reviewer's comments, and we have revised it according to the revision requirements. In the discussion section, we have clarified the concept and significance of the coordinated development of new urbanization and health production efficiency, and, we put forward recommendations related to the coordinated development of the two systems. 

6.Limited Discussion of Influencing Factors: 

The article mentions factors such as population density, economic development, government financial investment, and government health investment affecting coupling coordination. However, it does not elaborate on how these factors impact the relationship or their policy implications. 

Modification Description: We are very grateful for the reviewer's comments, and we have revised it according to the revision requirements. We have explained in more detail the influences of population density, economic development, government financial inputs, and government health inputs. 

7.Clarity and Organization:

The article would benefit from improved clarity and organization. The flow of information is somewhat fragmented, making it challenging for readers to follow the logical progression of the study. 

Modification Description: We are very grateful for the reviewer's comments, and we have revised it according to the revision requirements. In the abstract section, we have condensed its presentation, focusing on the significance, content, corresponding research methodology, and important research conclusions of this paper. In the introduction section, we have sorted out the logic of its paragraphs. The first paragraph introduces the interaction between health production efficiency and new urbanization, the second and third paragraphs are a review of the research on new urbanization and health production efficiency, respectively, and the last paragraph summarizes the research questions and research objectives of this paper. In the Materials and Methods section, we have merged the modeling and methodological explanations of new urbanization and health production efficiency, and merged the coupled coordination model and the relative development model, making this section clearer. In the results section, we have added a quantitative description of the two systems and their coupled coordination and given a relatively reasonable explanation for it. In the discussion section, we mainly discuss the significance of the coupled coordination of the two systems and give relevant suggestions. We also point out the limitations of this paper. The conclusion section condenses and summarizes the results. We have revised it accordingly, please see it in the revised manuscript. 

Kind regards,

Fang, Zhang

E-mail:2021212840@nwnu.edu.cn

---

## [Decision Letter · Decision Letter 1]

30 Jan 2024

The coupling coordination characteristics of China’s health production efficiency and new urbanization and its influencing factors

PONE-D-23-24900R1

Dear Dr. Zhang,

We’re pleased to inform you that your manuscript has been judged scientifically suitable for publication and will be formally accepted for publication once it meets all outstanding technical requirements.

Kind regards,

Yu Zhou

Academic Editor

PLOS ONE

Additional Editor Comments (optional):

Reviewers' comments:

Reviewer's Responses to Questions

**Comments to the Author**

1. If the authors have adequately addressed your comments raised in a previous round of review and you feel that this manuscript is now acceptable for publication, you may indicate that here to bypass the “Comments to the Author” section, enter your conflict of interest statement in the “Confidential to Editor” section, and submit your "Accept" recommendation.

Reviewer #2: All comments have been addressed

2. Is the manuscript technically sound, and do the data support the conclusions?

Reviewer #2: Yes

3. Has the statistical analysis been performed appropriately and rigorously? 

Reviewer #2: Yes

4. Have the authors made all data underlying the findings in their manuscript fully available?

Reviewer #2: Yes

5. Is the manuscript presented in an intelligible fashion and written in standard English?

Reviewer #2: Yes

6. Review Comments to the Author

Reviewer #2: All comments have been addressed. The paper is publishable in its current form. All comments have been addressed. The paper is publishable in its current form. All comments have been addressed. The paper is publishable in its current form.

7. PLOS authors have the option to publish the peer review history of their article (what does this mean?). If published, this will include your full peer review and any attached files.

Reviewer #2: No

---

## [Editor Report · Acceptance letter]

3 Mar 2024

PONE-D-23-24900R1 

PLOS ONE

Dear Dr. Zhang, 

I'm pleased to inform you that your manuscript has been deemed suitable for publication in PLOS ONE. Congratulations! Your manuscript is now being handed over to our production team.

Kind regards, 

on behalf of

Dr. Yu Zhou 

Academic Editor

PLOS ONE